

# Source and flux of POC in a karstic area in the Changjiang River watershed: impacts of reservoirs and extreme drought

Hongbing. Ji[1,2], Cai. Li[1], Huaijian Ding[1], Yang Gao[1]

[1]School of Civil & Environmental Engineering, University of Science and Technology Beijing, 100083 Beijing, China
[2]State Key Laboratory of Environmental Geochemistry, Institute of Geochemistry, Chinese Academy of Sciences, Guiyang 550002, China

*Correspondence to*: Hongbing Ji (hongbing_ji@163.com)

**Abstract.** Isotopes of carbon and nitrogen along with C/N ratios of particulate organic carbon (POC) were used to identify source and transformation of organic carbon in the suspended and surface sediments in a typical karstic watershed (Wujiang

River, an important tributary of the Changjiang River). Isotopic values indicated that POC was mainly derived from C3-dominated soil with increased contribution of phytoplankton in sites directly affected by reservoir. In contrast, the POC in surface sediments was mainly derived from C3- and C4-dominated soil with little reservoir influence. The positive correlations of carbon and nitrogen isotopes between suspended and surface sediments indicated that these two carbon pools were tightly coupled. On basis of the conservative estimation, POC transported $1.17 \times 10^{10}$ g to the Three Gorges Reservoir

in 2013. POC yield in Wujiang River ($0.13$ t km$^{-2}$ yr$^{-1}$) was much lower than large rivers with high carbonate percentage. Based on the carbonate distribution patterns of POC yield, percentage of carbonate area might not be a significant factor of riverine POC transport. The cascade of reservoir and extreme drought had significant influence on the POC flux in Wujiang River.

## 1 Introduction

Globally, about 0.4 Gt of riverine organic carbon is transported to the world's oceans each year, of which 0.15−0.17 Gt is POC (Hedges et al., 1997; Ludwig et al., 1996; Schlesinger and Melack, 1981). Rivers are important channels for transporting organic carbon from land to ocean. The riverine POC provides integrated information both on the natural processes and human activities within the drainage basin (Meybeck and Ragu, 1996; Kendall et al., 2001). Previous studies indicate that climate and human disturbance are two important factors of POC transport. For example, POC fluxes decrease

in response to dam construction and extreme drought (Bianchi et al., 2007; Yu et al., 2011; Zhang et al., 2009) and increase in response to deforestation and flood event (Kao and Liu, 1996; Sun et al., 2007).

Riverine POC ultimately originates from terrestrial and aquatic organic matter. Because different sources of POC are characterized by distinguished $\delta^{13}C_{POC}$, $\delta^{15}N_{TN}$ and C/N ratios, these indicators have been widely used to constrain sources and transformation of riverine POC. However, some physical and chemical process can alter the original composition of

element and isotopes, which should be carefully considered. For example, decomposition of organic matter decreases C/N ratios (Tremblay and Benner, 2006) and abundant inorganic nitrogen also limit the usefulness of C/N ratios as a tracer of



particulate organic matter source (Guerra et al., 2013; Hu et al., 2006). Therefore, researchers utilize different combinations of $\delta^{13}C_{POC}$, $\delta^{15}N_{TN}$ and C/N ratios to identify the sources and calculate associated contribution of these sources. For example, Kendall et al. (2011) use C/N ratios and $\delta^{15}N_{TN}$ as source criteria in 54% of the samples in four large river systems across the United States, while Wu et al. (2007) use $\delta^{13}C_{POC}$ and $\delta^{15}N_{TN}$ to estimate the contribution of POC sources in the Changjiang

River.

The Wujiang River drains a typical karst catchment with cascade of reservoir along the main stream. Numerous studies on POC have focused on global large rivers (Aucour et al., 2006; Bianchi et al., 2007; Coynel et al., 2005; Kendall et al., 2001; Ludwig et al., 1996; Meybeck and Ragu, 1996; Moreira-Turcq et al., 2003; Wang et al., 2012). However, little attention is paid to rivers draining karstic area, which are more subject to different geochemical processes and anthropogenic

activities than non-karst rivers (Liu, 2007; Ogrinc et al., 2008; Tao et al., 2009). Wujiang River flows into the Three Gorges Reservoir in Chongqing Municipality. However, related study on POC in the Wujiang River is still scarce after the Three Gorges Dam began impounding sediment in 2004. Based on analyses of $\delta^{13}C_{POC}$, $\delta^{15}N_{TN}$ and C/N ratios in the suspended and surface sediments, this study identified source and flux of POC in the Wujiang River and examined the impacts of reservoir and climate.

## 2 Materials and Methods

### 2.1 Study sites description

The Wujiang River is the largest tributary of the upper Changjiang River in its south bank. It originates from the eastern Wumeng Ranges in Yunnan-Guizhou Plateau and flows through Yunnan Province, Guizhou Province, Sichuan Province and Hubei Province. It flows into the Three Gorges Reservoir in Fuling District, Chongqing Municipality. The drainage area of

Wujiang River is 87920 km$^2$, of which 76% drains in Guizhou Province. Its average annual discharge is $534 \times 10^8$ m$^3$ yr$^{-1}$. The watershed belongs to a warm subtropical climate. The mean annual air temperature varies from 10°C to 18°C with mean annual precipitation from 800 mm to 1400 mm. The altitudes of Wujiang River range from 500 m to 2400 m with a decreasing trend from west to east. Plant species are diverse with the upper reach dominated by broadleaf evergreen forest and dryland crop, the middle reach dominated by evergreen broadleaf forest and deciduous broadleaf mixed forest, and the

lower reach dominated by subtropical evergreen Castanopsis forest. Yellow soil and limestone soil are dominant in the watershed (Zhang et al., 1995). The land use is dominated by forest land, cultivated land and grass land, which account for 50%, 31% and 18%, respectively. The soil erosion rate decreased from 2678 t km$^{-2}$ a$^{-1}$ in 1980s to 2313 t km$^{-2}$ a$^{-1}$ in 1990s due to sustainable soil conservation measures (Wang, 2011).

The Wujiang River is a typical karst watershed. In the upper reaches, Permian and Triassic carbonate rocks and basalt

are dominant with coal-bearing formations outcropping in the western area. In the midstream area, Permian and Triassic limestones, dolomitic limestones, and dolomites are dominant whereas the lower reaches are dominated by carbonate rocks intercalated with shales, sandy shales, and siltstones (Zhang et al., 1995).





## 2.2 Sampling and analyses

Twenty-five samples of suspended particulate matter (SPM) were collected from the mainstream and major tributaries of the Wujiang River (Fig. 1) in May and August 2013, respectively. Eighteen surface sediments were sampled in August 2013. An extreme drought occurred in most Guizhou Province during June and August. Fig. 2 showed the monthly water discharge and suspended sediment load in 2013 at the Wulong Hydrological Station, located in Wujiang River mouth. The data were taken from Changjiang Sediment Bulletin (2013). As plotted in the Fig. 2, water discharge and suspended sediment load decreased abruptly from June and August due to the drought event.

Sampling of SPM were conducted by filtration through precombusted (450℃ for 6h) and preweighted 47mm glass fiber filters for SPM weight concentrations and stable isotopic analyses of carbon and nitrogen. Surface sediments were collected using a 0.05 m$^2$ Van Veen grab (Jiang and Ji, 2013). All samples were stored in a freezer (<−20℃) prior to laboratory analyses.

The filter samples were freeze-dried before the particulate substance was scraped from the filter. The freeze-dried suspended matter and sediment were sieved to 200 mesh, treated with 1 M HCl to remove inorganic carbon, while nitrogen isotopes of particulate matter were measured on the bulk samples without acidification. Organic carbon (OC) and total nitrogen (TN) contents as well as isotopes of carbon and nitrogen were determined by using an elemental analyzer (Flash EA 1112HT, Thermo Fisher Scientific, Inc., USA) coupled with an isotope-ratio mass spectrometer (Finnigan Delta V Advantage, Thermo Fisher Scientific, Inc.) in the Laboratory of Stable Isotope Ratio Mass Spectrometry, Chinese Academy of Forestry (Beijing 100091, China). Stable isotope ratios are reported in δ−unit notation as follows:

$$\delta X(‰) = (R_{sample}/R_{standard} − 1) \times 1000 \tag{1}$$

where $R_{sample}$ is the $\delta^{13}C/^{12}C$ or $^{15}N/^{14}N$ ratios of the sample, and $R_{standard}$ is the corresponding ratios of sample standard. $\delta^{13}C$ values are reported relative to Pee Dee Belemnite (PDB) and $\delta^{15}N$ values are reported relative to $N_2$ in atmospheric air (AIR). Precision for $\delta^{13}C$ is 0.2‰ and for $\delta^{15}N$ is 0.2‰.

## 3 Results

### 3.1 The mineral properties of suspended particulate and surface sediments

The minerals of suspended particulate and surface sediment were analyzed in State Key Laboratory for Advanced Metals and Materials, University of Science and Technology, Beijing. As presented in the Fig. S1, the major minerals of SPM in the Wujiang River included detrital minerals (quartz, calcite and dolomite), clay minerals (illite, kaolinite and smectite), magnetite and ilmenite. In contrast, the surface sediments contained little clay minerals, reflecting the preferential enrichment of clay minerals in SPM. The enrichment of clastic carbonate minerals in surface sediment indicated the preferential sedimentation of clastic carbonate relative to clay minerals. This was similar to the study in the Yangtze River by Ding et al. (2014). The dolomite in the SPM and surface sediment was observed in the middle and lower reaches, which



was in agreement with the catchment lithology. The mineral composition of SPM and surface sediment reflected the process of physical and chemical weathering.

## 3.2 Elemental and isotope composition (carbon and nitrogen) in SPM

The ratios of carbon to nitrogen showed a wide range of 2.8−29.3, with a mean value of 13.6 in May and 8.8 in August
(Table S1), indicating the source with high C/N ratios in May and low C/N ratios in August. Fig. 3 showed the spatial and seasonal variations of C/N ratios, $\delta^{13}C_{POC}$ and $\delta^{15}C_{TN}$ in the Wujiang River. Compared with August, more samples in May had C/N ratios higher than 15. While more samples had C/N ratios <8 in August, which were distributed in sites near or in reservoirs, for example sites 1, 2 and 19 (Fig. 3). Considering the cascade of reservoirs in along the Wujiang River, the impact of reservoir should be examined. The sampling sites were divided into two kinds based on the relation with reservoirs:
sites directly affected by reservoirs and less affected by reservoirs. Table 1 showed the comparison of elemental and isotopic parameters in these two kinds of sites. As shown in the Table 1, C/N ratios were lower in sites directly affected by reservoir than those in sites less affected by reservoir.

$\delta^{13}C_{POC}$ of SPM in May ranged from −30.18 to −21.09‰ and averaged −26.30‰. Differently, $\delta^{13}C_{POC}$ in August displayed relatively depleted values with an average value of −27.23‰. $\delta^{15}N_{TN}$ of SPM ranged from 1.88‰ to 12.93‰ and
averaged 6.82‰, with higher values in August (7.58‰) than those in May (6.05‰). Spatially, $\delta^{13}C_{POC}$ values, especially in August, were more depleted in reservoirs and sites downstream of the reservoirs (−28.65±1.22‰, Table 1 and Fig. 3) than those less affected by reservoirs (−26.68±3.23‰, Table 1 and Fig. 3). In contrast, $\delta^{15}N_{TN}$ values were more enriched in sites directly affected by reservoirs than other sites. Chen and Jia (2009) obtained the similar trend of $\delta^{13}C_{POC}$ and $\delta^{15}N_{TN}$ in a dam-controlled subtropical river. As presented by Fig. 3, the heavier $\delta^{15}N_{TN}$ values (>10‰) in August corresponded to
lighter $\delta^{13}C_{POC}$ values and lower C/N ratios (Fig. 3).

To test the relationships between TSS concentrations, elemental compositions (POC%, TN% and C/N ratios) and isotopic compositions, Person's correlation coefficient (Table S2) was calculated using SPSS software (version 16.0 for Windows). As shown in Table S2, POC%, TN% and $\delta^{15}N_{TN}$ were significantly negatively correlated with TSS concentrations in May, i.e., POC%, TN% and $\delta^{15}N_{TN}$ decreased when TSS concentrations increased. In contrast, $\delta^{13}C_{POC}$ was
significantly positively correlated with TSS concentrations, i.e., $\delta^{13}C_{POC}$ increased when TSS concentrations increased. Similar results are obtained in the freshwater part of the Scheldt Estuary (Hellings et al., 1999). However, the relationships between isotopic parameters and TSS concentrations in August were not significant. This might be due to the low concentrations of TSS in August, when extreme drought occurred in most Guizhou Province. No significant correlation of $\delta^{15}N_{TN}$ and $\delta^{13}C_{POC}$ was observed, which could be related to the inorganic nitrogen in samples (Guerra et al., 2013; Hu et al.,
30 2006).



### 3.3 Compositions of element and isotopes (carbon and nitrogen) in surface sediments

Carbon to nitrogen ratios in the surface sediments presented higher values ranging from 7.7 to 41.1 compared with those of SPM. Like the SPM, POC% and TN% in surface sediments displayed a relatively significant positive correlation (R=0.595, P<0.01), which indicated that some nitrogen in the sediments was inorganic nitrogen. Compared with the C/N ratios in SPM,

surface sediments had higher C/N ratios, indicating different sources of POC in surface sediments.

$\delta^{13}C_{POC}$ of surface sediments showed a relatively narrow range of −26.40‰ to −22.73‰ with an average value of −24.76‰. The enriched $\delta^{13}C_{POC}$ of surface sediments in contrast with SPM indicated a source with elevated $\delta^{13}C_{POC}$ values. The variation trend of $\delta^{13}C_{POC}$ in surface sediments was similar to that in SPM, which indicated these two carbon pools might be coupled. $\delta^{15}N_{TN}$ of surface sediments ranged from 2.88‰ to 9.39‰ with a mean value of 6.01‰. Spatially, the

mean values of $\delta^{13}C_{POC}$ and $\delta^{15}N_{TN}$ in sites directly affected by reservoir and sites less affected by reservoir were −24.85±0.91‰ vs. −24.73±1.04‰ and 6.23±2.22‰ vs.5.93±1.55‰ (Table 1), respectively. Compared with the SPM, the difference of isotopic values were smaller in surface sediment between sites directly by reservoir and sites less affected by reservoir.

## 4 Discussion

### 4.1 Sources and variations of POC in the Wujiang River

### 4.1.1 Sources of POC in SPM

Potential sources of POC in the river contain allochthonous sources (C3 and C4 plants, soil organic matter) and autochthonous sources (macrophytes and phytoplankton). C/N ratios and isotopic values of POC are an effective method for constraining the sources of riverine POC. Generally, aquatic phytoplankton is charactered with low C/N ratios (5−8) and

terrestrial organic matter with high C/N ratios (higher than 8, Kendall et al., 2001). Soil organic matter reflects carbon isotopic compositions of residues from the overlying vegetation with an average $\delta^{13}C$ value of −27.0‰ for C3 plants and −14‰ for C4 plants (Smith and Epstein, 1971). The typical $\delta^{15}N$ values for soil organic nitrogen are 2−5‰ (Kendall et al., 2001).

C/N ratios as well as isotopic compositions of carbon are plotted in Fig. 4 together with typical values of potential end

members of POC. As shown in the Fig. 4, C/N ratios of SPM in the Wujiang River showed temporal variations with higher C/N ratios in May (13.6±7.6) than those in August (8.8±3.7), which suggested the dominant terrestrial contribution to SPM in May and increased phytoplankton input in August. Of note, the precondition of C/N ratios for identifying organic matter sources was that all of TN in POC exclusively reflected nitrogen bound to organic matter (Meyers, 1997). Therefore, contents of organic carbon (POC%) and total nitrogen (TN%) was expected to show a significant linear correlation. The

linear relationship of TN and POC was relatively weak (May: TN=0.07*POC+0.09, $R^2$=0.54, P<0.001; August: TN=0.04*POC+0.23, $R^2$=0.39, P<0.001) compared with other studies ($R^2$=0.71 in Hu et al., 2006; $R^2$=0.9 in Guerra et al.,



2013). The intercept of the above regressions was more than zero, which suggested that a fraction of TN was inorganic nitrogen in the SPM (Guerra et al., 2013; Hu et al., 2006). Thus, the phytoplankton inputs might be overestimated based on C/N ratios.

$\delta^{13}C_{POC}$ of SPM in May and August averaged −26.30‰ and −27.23‰, respectively. The depleted $\delta^{13}C_{POC}$ in August

indicated decreased terrestrial contribution and increased phytoplankton contribution. This was in accordance with the conclusion deduced from the C/N ratios. This was similar to other rivers with large carbonate area, for example Sava River (KanduČ et al., 2007; Ogrinc et al., 2008) and Xijiang (Sun et al., 2007), where POC were mainly derived from terrestrial matter. The Wujiang River had high flow rates and rocky river beds and banks, which limited the macrophytes growth. Thus, phytoplankton was the main aquatic plants in the catchment (Tao et al., 2009). Phytoplankton was reported to have depleted

$\delta^{13}C$ values with a typical range of −42‰ to −24‰ (Kendall et al., 2001 and references therein). The $\delta^{13}C_{POC}$ values of SPM were more negative in reservoir-affected sites than those unaffected by reservoir (Table 1). This could be due to long water retention time of the reservoir, which was in favor of phytoplankton enhancement. The aquatic source increase in the reservoir was reported in other rivers (Chen and Jia, 2009; Zhang et al., 2009).

The contribution proportions were calculated by a mixing model based on the $\delta^{13}C_{POC}$ values. The $\delta^{13}C$ of

phytoplankton end member can be estimated based on the measured $\delta^{13}C$ values of dissolve inorganic carbon ($\delta^{13}C_{DIC}$) and an uptake fractionation of 21‰ (i.e. $\delta^{13}C$ of phytoplankton=$\delta^{13}C_{DIC}$−21‰, Mook and Tan, 1991). Measured $\delta^{13}C$-DIC in the Wujiang River ranged from −11.55‰ to −3.41‰, with an average value of -8.67‰ (unpublished data). Thus, the estimated $\delta^{13}C$ values for phytoplankton ranged from −32.55‰ to −24.41‰ with an average value of −29.67‰. This results fell in the typical $\delta^{13}C$ range (−42‰ to −24‰) of freshwater plankton (Kendall et al., 2001 and references therein). This was

also in accordance with the study by Li (2009) in Maotiao River (a tributary of Wujiang River), where the average of $\delta^{13}C_{POC}$ of phytoplankton was −29.6±5.5‰. Therefore, the depleted $\delta^{13}C_{POC}$ values (<−29.6‰) in the catchment reflected the dominated phytoplankton inputs. The Wujiang River is the largest tributary of the upper Changjiang River in its south bank. Wu et al. (2007) reported the average $\delta^{13}C$ value of soil end-member (−26.1±0.3‰) within the southern tributaries of the upper Changjiang River, which was taken as the upper limit of C3 plant-dominated soil end-member. That is, if −29.6‰<

$\delta^{13}C_{POC}$ <−26.1‰, the POC was derived from the mixing of phytoplankton and C3 plant-dominated soil. The corresponding mixing model is:

$$\delta^{13}C_{POC} = \delta^{13}C_{POC-phyto} \times f_{phyto} + \delta^{13}C_{POC-C3} \times (1 - f_{phyto}) \qquad (2)$$

where $\delta^{13}C_{POC}$, $\delta^{13}C_{POC-phyto}$ and $\delta^{13}C_{POC-C3}$ are the $\delta^{13}C$ values of collected samples, phytoplankton and C3 plant-dominated soil, respectively and $f_{phyto}$ is the contribution proportion of phytoplankton. The average $\delta^{13}C$ of C4 plants in the catchment

was −13.4‰ (Tao et al., 2009), which was taken as the upper limit of C4 plant sources. That is, if −26.1‰< $\delta^{13}C$ <−13.4‰, the POC was derived from the mixing of C3 plant-dominated soil and C4 plant-dominated soil. The corresponding mixing equation is:

$$\delta^{13}C_{POC} = \delta^{13}C_{POC-C3} \times f_{C3} + \delta^{13}C_{POC-C4} \times (1 - f_{C3}) \qquad (3)$$



where $\delta^{13}C_{POC}$, $\delta^{13}C_{POC-C3}$ and $\delta^{13}C_{POC-C4}$ are the $\delta^{13}C$ values of collected samples, C3 plant-dominated soil and C4 plant-dominated soil, respectively and $f_{C3}$ is the contribution proportion of C3 plant-dominated soil. The calculated results are presented in Fig. 5. POC sources of SPM varied temporally with C3 accounting for 76% in May and decreasing significantly to 48% in August. Phytoplankton contributions increased from 18% in May to 47% in August. The average phytoplankton

contribution in sites directly affected by reservoir was 47%, higher than other sites with an average phytoplankton contribution of 27%.

### 4.1.2 Sources of POC in sediments

C/N ratios and the $\delta^{13}C_{POC}$ values of surface sediments reflected the compositions of organic matter in recent decades (Krull et a., 2009). C/N ratios of surface sediments ranged from 7.7 to 41.1 with an average value of 18.0, which verified the

dominant terrestrial sources. Compared with SPM, the elevated C/N ratios of surface sediments indicated more land-derived fraction contribution to the surface sediments. The POC% and TN% in surface sediments exhibited relatively strong correlation (R=0.595, P<0.01), which suggested that there was some inorganic nitrogen in the surface sediments. From June to August 2013, drought hit most parts of Guizhou Province. Soil organic matter and plant debris might be deposited and mineralized on the ground before they were transported into the rivers. Consequently, contents of organic components

decreased and inorganic components increased, which might result in the weak correlation between POC% and TN%.

The relation of $\delta^{13}C_{POC}$ and C/N in surface sediments (Fig. 4b) indicated that POC was mainly derived from terrestrial origin. In contrast with the SPM, the enriched $\delta^{13}C_{POC}$ values of surface sediments (averaging −24.76‰) suggested that there was an increased source of C4 plants to sediments, and/or depleted carbon isotope in surface sediment were not retained in the sediment (Guerra et al., 2013). Given that POC and TN contents were higher in most sediment samples than suspended

sediments, the biodegradation of the phytoplankton was not the major cause of enrichment of $\delta^{13}C_{POC}$ values in surface sediments.

The contributions of different sources were calculated based on the equations (2) and (3). Contribution of C3 plant-dominated soil and C4 plant-dominated soil averaged 89% and 11%, respectively. There was almost no phytoplankton input except for one site (Fig. 5). Spatial variation of source contribution was not significant in the surface sediment (Fig. 5),

indicating that reservoir influence was relatively weak in surface sediment compared with those in SPM.

### 4.2 Transformation of POC in the Wujiang River

Knowledge of the POC transformation is useful to get a better understanding of the riverine carbon cycle. In-stream processes, such as assimilation and respiration of phytoplankton, affect the isotopic compositions and element contents of carbon and nitrogen. Hence, $\delta^{13}C_{POC}$、$\delta^{15}N_{TN}$ and C/N can be utilized to trace transformation processes of organic matter.

The trend of increasing TSS concentrations with decreasing POC contents (%, Fig. 6) indicated that POC contents (%) of SPM were diluted with the inorganic constituents derived from soil erosion. This was similar to other rivers (Ludwig et al., 1996; Jiang and Ji, 2013; Zhang et al., 2009). The positive correlation between TSS concentrations and POC contents (μmol



L$^{-1}$, Table S2) indicated that terrestrial organic matter was an important source of POC in SPM, which confirmed the erosion process. Although the erosion rate catchment decreased significantly in the Wujiang River (Wang, 2011), soil erosion had remarkable influence on the riverine carbon cycles.

$\delta^{15}$N is a potential tracer to identify aquatic biogeochemical processes. High $\delta^{15}$N may be caused by anthropogenic

activities and transformation processes, such as denitrification and assimilation. Denitrification was excluded due to the weak correlation between $\delta^{15}$N-NO$_3^-$ and $\delta^{18}$O-NO$_3^-$ (unpublished data). Kendall et al. (2007) reported that animal waste and domestic effluents had typical values of $\delta^{15}$N-NO$_3^-$ >10‰. Some samples had high $\delta^{15}$N$_{TN}$ values with elevated $\delta^{15}$N-NO$_3^-$ values (>8%), indicating the inputs of manure and domestic sewage. Particulate organic carbon could be influenced by sewage water through the uptake of NH$_4^+$ and NO$_3^-$ by phytoplankton. Uptake of NO$_3^-$ with high $\delta^{15}$N by phytoplankton

might result in elevated $\delta^{15}$N$_{TN}$ values (Kendall et al., 2001; Jiang and Ji, 2013). This process was confirmed by the significant positive correlation of $\delta^{15}$N$_{TN}$-SPM vs. $\delta^{15}$N-NO$_3^-$ and $\delta^{15}$N$_{TN}$-SPM vs. NO$_3^-$ in May (Fig. 7a). However, some samples with $\delta^{15}$N$_{TN}$-SPM >10‰ ($\delta^{15}$N-NO$_3^-$ <8‰) (Fig. 7a) deviated from other data in August, indicating other influencing factors. One mechanism might be related to uptake of other forms of dissolved inorganic nitrogen. The drought event during June to August created longer time for mineralization of soil organic matter, favourable to the production of

ammonia with heavy $\delta^{15}$N values. The elevated $\delta^{15}$N$_{TN}$-SPM might be caused by the uptake of nitrification-derived NH$_4^+$, as nitrifiers preferentially consumed $^{14}$N, leading to increase in $\delta^{15}$N-NH$_4^+$ of the remaining ammonia. Consumption of such enriched $^{15}$N-NH$_4^+$ by phytoplankton resulted in the scattered trend of $\delta^{15}$N$_{TN}$-SPM in August. The similar result was observed in the study by Sarma et al. (2012).

Positive correlations of $\delta^{13}$C$_{POC}$ (Fig. 7b) and $\delta^{15}$N$_{TN}$ (Fig. 7c) between suspended and surface sediments indicated that

intense exchange might exist in these two carbon pools (Jiang and Ji, 2013; Sarma et al., 2012). The resuspension/deposition of suspended matter with mixing of different organic matter sources might result in the significant correlation of $\delta^{13}$C$_{POC}$ (Fig. 7b) and $\delta^{15}$N$_{TN}$ (Fig. 7c). Compared with suspended matter, the heavier $\delta^{13}$C$_{POC}$ and lighter $\delta^{15}$N$_{TN}$ in sediments indicated enriched source of refractory allochthonous organic matter. The good correlations between river water and SPM as well as surface sediment indicated that these carbon pools were tightly connected.

**4.3 Flux of POC in Wujiang River and comparison with world rivers**

Flux of POC (F$_{POC}$) was estimated based on a simple method from Tao et al. (2009):

F$_{POC}$ = [POC]$_H$ × Discharge × 66% + [POC]$_L$ × Discharge × 34% (4)

where [POC]$_H$ and [POC]$_L$ is the average POC concentration of SPM in high-water season and low-water season, respectively. The POC concentration of river mouth (sample 18 in Fig. 1) was used to calculate the POC flux. Water

discharge of the Wujiang River in 2013 (website: www.cjw.gov.cn) was taken from the Wulong hydrologic station, which was close to the river mouth. Discharge in high-flow and low-flow season account for 66% and 34% of the annual discharge in 2013, respectively (Changjiang Sediment Bulletin 2013). Since the water discharge in May was the almost highest compared with other months (Fig. 2), the POC concentration in May collected at river mouth (site 18 in Fig. 1) was used to





calculate the POC flux in high-water season. Similarly, POC concentration in August was used to calculate the POC flux in low-water season since water discharge in August was close to that in low-water season (Fig. 2). The estimated POC flux was $1.17 \times 10^{10}$ g in 2013, lower than that in the Wujiang River determined by Tao et al. (2009) in 2002. The decrease in POC flux might be due to measures of soil and water conservation, dam construction (Wu et al., 2007) as well as decreased

discharge due to extreme drought in 2013 in Guizhou Province.

In Table 2 water discharge, TSS concentration, POC%, POC flux and yield were compared with 15 world rivers. The total drainage area of the 15 rivers amounted to $30.3 \times 10^6$ km$^2$ (Table 2), accounting for 55% of the 60 world rivers' area from the study of Gaillardet et al. (1999). As shown in Table 2, the very low POC yield in Wujiang River was the forth lowest observed in the documented rivers before Yenisey, Ob and St. Lawrence. In comparison, the first two rivers in terms

of POC yield were Zhujiang and Ganges-Brahmaputra with POC yield higher than 3 t km$^{-2}$ yr$^{-1}$. Both Zhujiang and Ganges-Brahmaputra were located in the mid latitude, subject to tropical and subtropical climate. The latitudinal distribution patterns of POC yield were examined for the 15 world rivers and Wujiang River (Fig. 8a). As shown in the Fig. 8a, POC yields were higher in the mid-latitude rivers and tended toward the subtropical rivers. This was similar to the distribution feature of HCO$_3^-$ yield for the world rivers in the study of Cai et al. (2008). According to Amiotte Suchet et al. (2003), carbonate rocks

were mainly distributed between 20°N and 50°N. The carbonate area in the documented 15 world rivers accounted for 8% of the 60 world rivers' area, while POC flux accounted for 13% of the global POC flux of 0.17 Gt ($10^{15}$ g) estimated by Ludwig et al. (1996). It appeared that more carbonate would result in elevated POC export. Unfortunately, no clear correlation was found between carbonate percentage and POC yields (Fig. 8b), which indicated that percentage of carbonate area was not a significant factor of riverine POC transport. This was not similar to Mackenzie River, where organic-rich sedimentary rocks

contributed a significant particulate organic matter (Carrie et al., 2009). However, the influence of carbonate might be underestimated as discussed by Cai et al. (2008) since carbonate rocks were defined as those that contained up to 50% of carbonate minerals (Amiotte Suchet et al., 2003).

Compared with the previous study in Wujiang River, the POC yield decreased from 0.47 t km$^{-2}$ yr$^{-1}$ (Tao et al., 2009) to 0.13 t km$^{-2}$ yr$^{-1}$ (Table 2). It was noted that five dams were constructed in the lower reach of Wujiang River after the

study of Tao et al. (2009). Moreover, POC yield in the Wujiang River were much lower than those in Xijiang and Zhujiang with high carbonate percentage. This could be related to the smaller watershed area and extensive water reservoirs in the Wujiang River (Zhang et al., 2006). The impacts of reservoirs will be discussed below.

### 4.4 Impacts of reservoir and climate on riverine POC

Eleven artificial dams have been constructed along the mainstream of Wujiang River (Fig. 1) since 1970s. The cascade of

reservoirs created by dams may exert significant impacts on source and transport of riverine POC. In addition, extreme drought must be noted when considering the meteorological characteristics of the Wujiang River in 2013.

$\delta^{13}$C and $\delta^{15}$N proved to be potentially useful indicators for qualitatively estimation of reservoir and climatic impact on POC in dam-affected rivers (Chen and Jia, 2009; Zhang et al., 2009). In order to analyze the reservoir impact on riverine



POC, $\delta^{13}C$ and $\delta^{15}N$ values were compared in sites directly affected by dams with those less affected in the Wujiang River (Table 1). As shown in Table 1, $\delta^{13}C$ were more depleted in sites affected by dam than those less affected by dam. This was similar to the study by Chen and Jia (2009) in a dam-controlled river. Compared with sites far from the dam, the more depleted values of $\delta^{13}C$ in sites close to dam were attributed to increasing phytoplankton contribution. This was confirmed

by the higher phytoplankton contribution to POC of SPM in sites directly affected by dam with an average of 47% relative to those less affected with an average of 27%. Two mechanisms could explain this elevated phytoplankton contribution: (1) extended water retention time in reservoirs with low flow; (2) increasing light availability due to the low TSS concentrations (Table 1) in reservoirs. In contrast with $\delta^{13}C$ values, $\delta^{15}N$ values were heavier in sites close to reservoir than those far from reservoir. Delong and Thorp (2006) reported that $\delta^{15}N$ values in aquatic algal were more enriched than detrital fraction. Thus,

the heavier $\delta^{15}N$ values in sites directly affected by dam might result from the aquatic algal and plankton. This hypothesis was verified by the concurrent lighter $\delta^{13}C$ values and lower C/N ratios in the sites directly influenced by reservoirs (Table 1). Besides, denitrification could be another cause for the heavier $\delta^{15}N$ values in the reservoirs since hypoxic environments might occur in the deep water. Further study of $\delta^{15}N$ variations in different water depths could help to trace the denitrification process. Seasonally, the average $\delta^{13}C$ values in August at sites directly affected by reservoir (−28.65‰) were

much higher than other sites (−26.68‰). This large difference could be related to the higher temperature and extreme drought during June to August, which was favourable to in-situ phytoplankton production. According to Chen and Jia (2009), accumulated terrestrial organic matters in winter were flushed during the first heavy rain, which resulted in increasing terrestrial input in the onset of the wet season. This could also be the reason for the Wujiang River in May, when the water discharge increased abruptly.

It is difficult to quantitatively evaluate the impact of dams and extreme drought. The method described in the study of Yu et al. (2011) made it possible to distinguish the impact of dams and climate. This method was based on the significant correlation between POC flux and suspended sediment load. Thus, the variations of suspended sediment load could reflect the POC flux variations under the condition of dam and extreme drought. The comparison of suspended sediment loads was made between normal years and the drought year 2013 in the Wulong hydrologic station, the Wujiang River mouth (Table

S3). As shown in Table S3, the suspended sediment loads in 2013 at Wulong station reduced by 80% compared with normal years, which could be due to the combined impacts of cascade of dams and extreme drought. Because suspended sediment at Wulong station directly flowed into Three Gorges Dam (TGD), the impact of climate on TGD sediment revealed the similar impacts on Wulong station. The impacts of extreme drought can be estimated based on the comparison of sediments inputs to TGD between normal and drought year. The reduction 41% of sediment inputs to TGD was obtained in 2013 compared

with normal years, which was attributed to the climate impact. Thus, the impact of cascade of dams was 39%. The normalized impacts of cascade of dams and extreme drought were 49% and 51%, respectively. This result was similar to the impacts of extreme drought in 2006 on TGD determined by Yu et al. (2011). This result indicated that extreme drought and dams were important factors of suspended sediment load. Considering the significant correlation between TSS

concentrations and POC concentrations (Table S2), the reservoir and extreme drought had similar impact on the POC transport. However, this estimation based on the Wujiang River mouth was limited relative to the whole basin.

## 5 Conclusions

The carbon to nitrogen ratios and its isotopic compositions of POC were determined in suspended and surface sediments in Wujiang River. The results indicated that POC in SPM was mainly derived from C3-dominated soil with increased phytoplankton input in sites affected by reservoirs. In comparison, POC in surface sediments was mainly derived from C3- and C4-dominated soil. The relationships of carbon and nitrogen isotopes between suspended and surface sediments indicated that these two carbon pools are closely coupled. In-stream process, such as microbiological decomposition in water column and surface sediments, might result in the difference in terms of POC sources between suspended and surface sediments. POC transported $1.17 \times 10^{10}$ g to the Three Gorges Reservoir in 2013. POC yield in Wujiang River (0.13 t km$^{-2}$ yr$^{-1}$) was much lower than large rivers with high carbonate percentage. The carbonate distribution patterns of POC yield indicated that percentage of carbonate area was not a significant factor of riverine POC transport. The cascade of reservoir and extreme drought had significant influence on the POC flux in Wujiang River.

## Author contributions

Research plan: Hongbing Ji

Experimental design: Cai Li, Hongbing Ji

Financial support: Hongbing Ji

Methodology (XRD, isotopes, statistical analysis): Cai Li, Hongbing Ji

Experimental implementation and data analysis: Cai Li, Huaijian Ding, Yang Gao, Hongbing Ji

Manuscript drafting and reviewing: Cai Li, Hongbing Ji

Hydrological and carbonate chemistry data support: Huaijian Ding, Yang Gao

## Acknowledgments

We also thank Dr. Meryem Briki for her assistant with field sampling. Furthermore, we are indebted to Academician & Prof. Liu Congqiang and Prof. Wang Shijie for discussion and suggestions about this study. This work was financially supported by the National Natural Science Foundation of China (No. 41473122), the National Key Basic Research and Development Program (No. 2013CB956702) and the Hundred Talents Programs of Chinese Academy of Sciences. All data used in this publication will be provided upon request by the corresponding author.



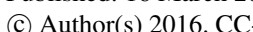

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



Table 1 Comparison of TSS concentrations, C/N ratios and isotopic values (mean ± standard deviation) in sites affected by reservoirs with those unaffected by reservoirs in the Wujiang River

| Sites | Descriptions | TSS (mg L$^{-1}$) | | C/N ratios | | | $\delta^{13}$C (‰) | | | $\delta^{15}$N (‰) | | |
|---|---|---|---|---|---|---|---|---|---|---|---|---|
| | | May | August | May | August | Sediment | May | August | Sediment | May | August | Sediment |
| 1, 2, 16, 19, 20, 23, 24 | reservoirs and downstream of the reservoirs | 5.39 ±6.99 | 3.79 ±1.58 | 13.1 ±8.9 | 5.9 ±2.4 | 18.0 ±6.9 | -26.77 ±1.44 | -28.65 ±1.22 | -24.85 ±0.91 | 6.82 ±1.97 | 7.99 ±4.12 | 6.23 ±2.22 |
| Others | less affected by reservoirs | 24.35 ±33.66 | 4.80 ±3.90 | 13.7 ±7.2 | 10.0 ±3.6 | 18.0 ±10.7 | -26.11 ±2.06 | -26.68 ±3.23 | -24.73 ±1.04 | 5.75 ±1.49 | 7.42 ±2.49 | 5.93 ±1.55 |





Table 2 Particulate organic carbon fluxes (FPOC) and yields (YPOC) in some World Rivers and Wujiang River

| River | Area 10³ km² | Latitude[1] | Discharge km³ yr⁻¹ | Carbonate[2] % | TSS mg L⁻¹ | POC % | FPOC Mt yr⁻¹ | YPOC t km⁻² yr⁻¹ | Source |
|---|---|---|---|---|---|---|---|---|---|
| Amazon | 5854 | 2 | 6642 | 3.9 | 54.7 | 8.62 | 5.00 | 0.85 | Moreira et al. (2003) |
| Changjiang | 1794 | 30 | 779 | 44 | 134 | 1.21 | 1.52 | 0.85 | Wang et al. (2012) |
| Congo/Zaire | 3699 | 4 | 1325 | 10.1 | 26 | 6.5 | 2.00 | 0.50 | Coynel et al. (2005) |
| Danube | 788 | 48 | 207 | 14.5 | 39 | 4.10 | 0.25 | 0.32 | Reschkee et al. (2002) |
| Ganges−Brahmaputra | 1648 | 26 | 1313 | 33.8 | 287 | 2.23 | 6.00 | 3.64 | Ancouret al. (2006) |
| Huanghe | 752 | 36 | 13 | 7.6 | 2522 | 0.48 | 0.39 | 0.52 | Wang et al. (2012) |
| Indus | 1143 | 29 | 104 | 26 | 1917 | 0.46 | 2.05 | 1.79 | Ludwig et al. (1996) |
| Lena | 2418 | 63 | 525 | 11.2 | 20 | 3.75 | 0.38 | 0.16 | Semiletov et al. (2011) |
| Mackenzie | 1713 | 64 | 290 | 20.6 | − | − | 1.10 | 0.64 | Carrie et al. (2009) |
| Mississippi | 3203 | 36 | 610 | 18.1 | 112 | 1.6 | 0.93 | 0.29 | Bianchi et al. (2007) |
| Ob | 2990 | 60 | 412 | 2.7 | − | − | 0.31 | 0.10 | Köhler et al. (2003) |
| St. Lawrence | 1267 | 47 | 363 | 24.9 | − | − | 0.13 | 0.10 | Helie (2004) |
| Wujiang River | 67 | 27 | 38 | 73.6[3] | 25.99 | 9.21 | 0.03 | 0.47 | Tao et al. (2009) |
| Wujiang River | 88 | 27 | 33 | 73.6[3] | 38 | 1.47 | 0.01 | 0.13 | This study |
| Xijiang | 353 | 23 | 182 | 82.4 | 232 | 1.5 | 0.43 | 1.21 | Sun et al. (2007) |
| Yenisey | 2580 | 60 | 599 | 6.9 | − | − | 0.17 | 0.07 | Köhler et al. (2003) |
| Zhujiang | 427 | 23 | 343 | 35 | − | − | 2.50 | 5.75 | Zhang et al. (2013) |
| Mean(M) or Sum(S)[4] | 30276(S) | 38(M) | 13525(S) | 18.5(M) | 568(M) | 3.22(M) | 22.73(S) | 1.11(M) | |

[1]Cai et al. (2008)

[2]Amiotte Suchet et al. (2003)

[3]Carbonate% in Wujiang was in reference to the carbonate distribution inside Guizhou Porvince (Wan, 1995)

[4]Xijiang and Wujiang, as tributaries of Zhujiang and Changjiang, respectively, were not included in the statistical data.





Fig. 1 Location of sampling sites in the Wujiang River. Zones of W I, W IIand WIII are the upper reaches, middle reaches

and lower reach, respectively. W I is dominated by Permian and Triassic carbonate rocks and basalt with coal-bearing

5   formations outcrop in west. W II is dominated by Permian and Triassic limestones, dolomitic limestones, and dolomites.

WIII is distributed carbonate rocks intercalated with detrital rocks (shales, sandy shales, and siltstones).



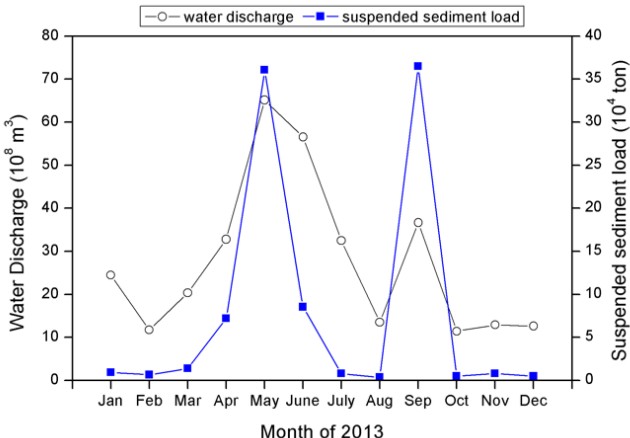

Fig. 2 Monthly water discharge and suspended sediment load in 2013 at Wulong Hydrological Station. The data were taken

5      from Changjiang Sediment Bulletin 2013 (website: www.cjw.gov.cn).





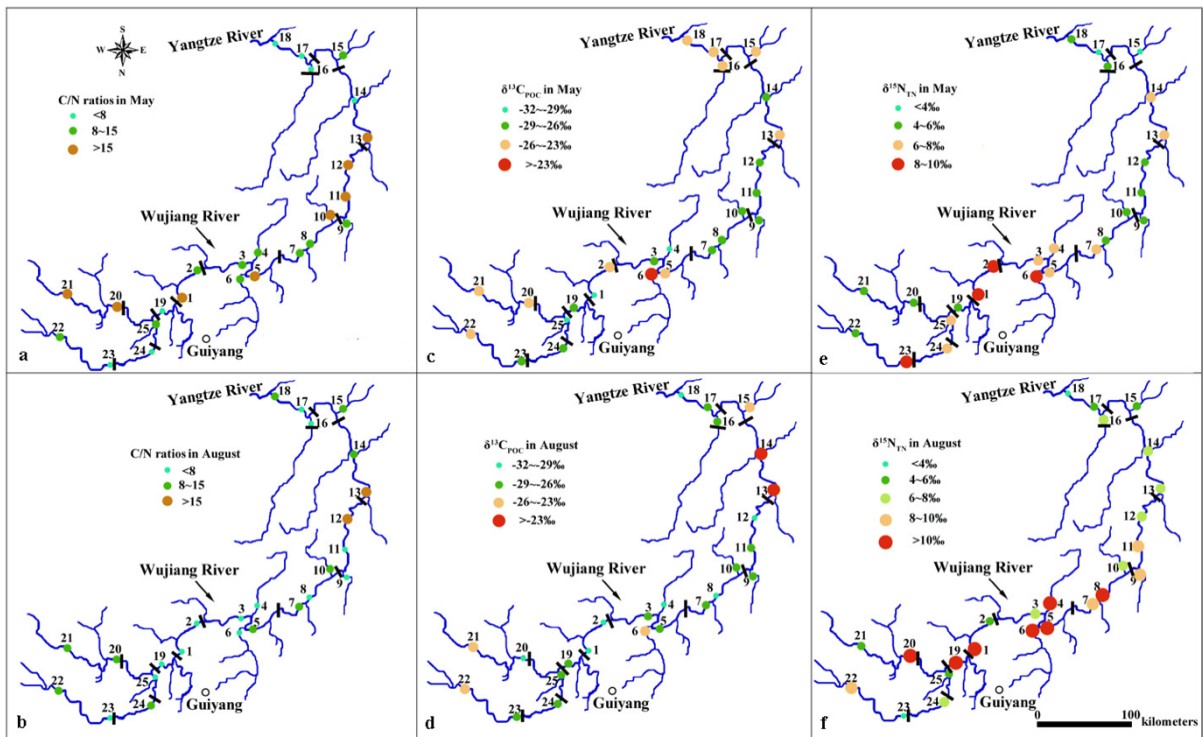

Fig. 3 The spatial variations of C/N ratios, $\delta^{13}C_{POC}$ and $\delta^{15}C_{TN}$ of suspended particulate matter in the Wujiang River.



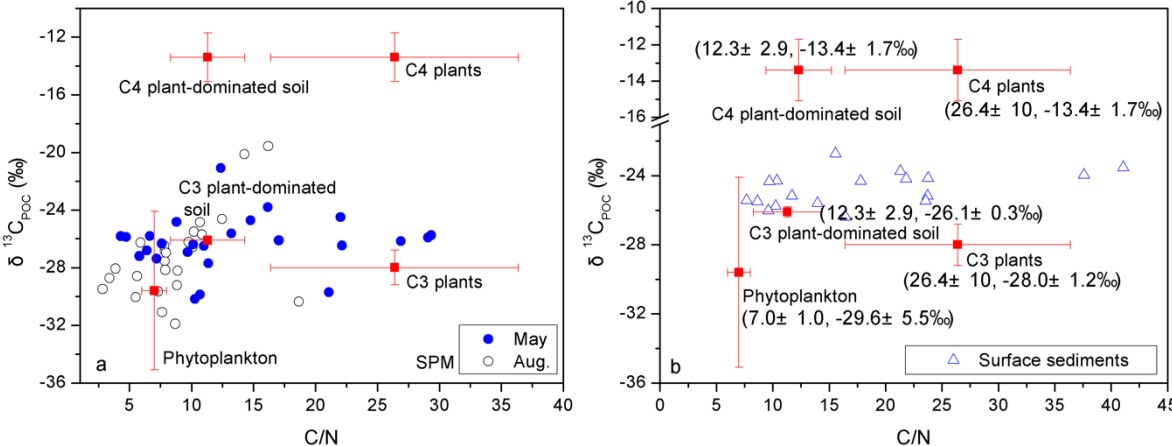

Fig. 4 $\delta^{13}C_{POC}$ and C/N ratios of suspended and surface sediments in the Wujiang River. The isotopic and elemental compositions of different end-members are taken from Li (2009), Wu et al. (2007) and Tao et al. (2009) and references therein.





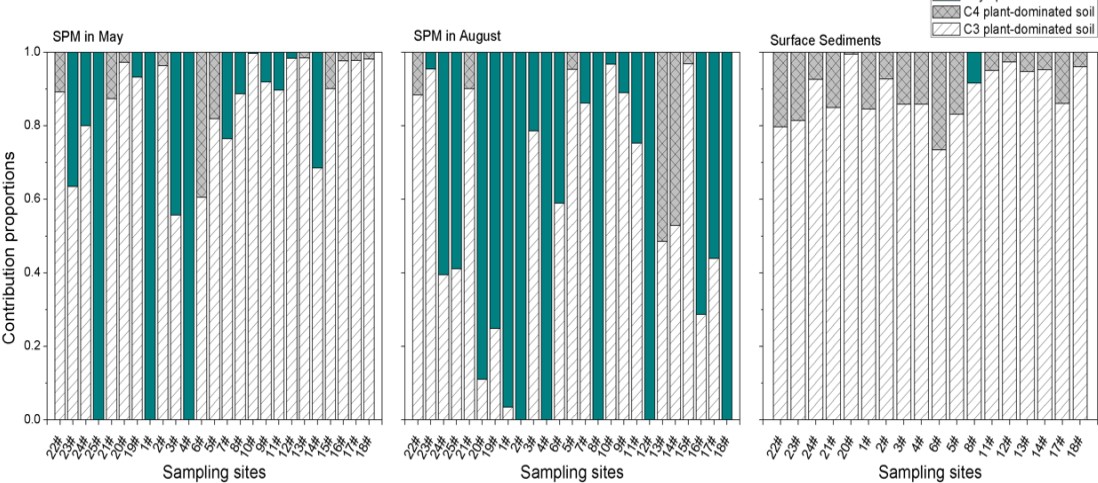

Fig. 5 Contribution proportions of different sources to POC in suspended matters and surface sediments.





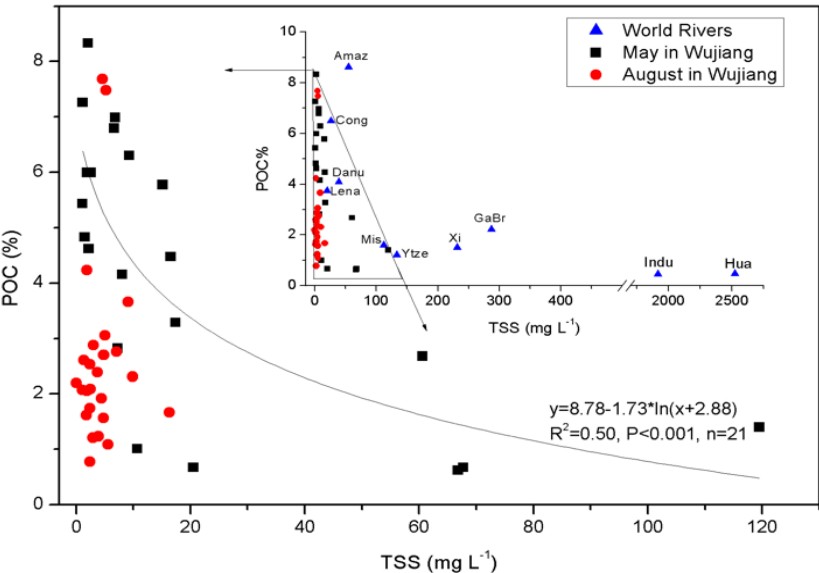

Fig. 6 Correlations between TSS concentrations (mg L$^{-1}$) and POC (%).The related data were summarized in Table 2.
Amazon, Changjiang, Congo, Danube, Ganges-Brahmaputra, Huanghe, Indus, Lena, Mackenzie, Mississippi, St.
Lawrence, Wujiang, Xijiang, Yenisey, Zhujiang are abbreviated to Amaz, Chang, Cong, Danu, Gabr, Hua, Inds, Lena,
Mack, Mis, StL, Wu, Xi, Yens, Zhu, respectively.



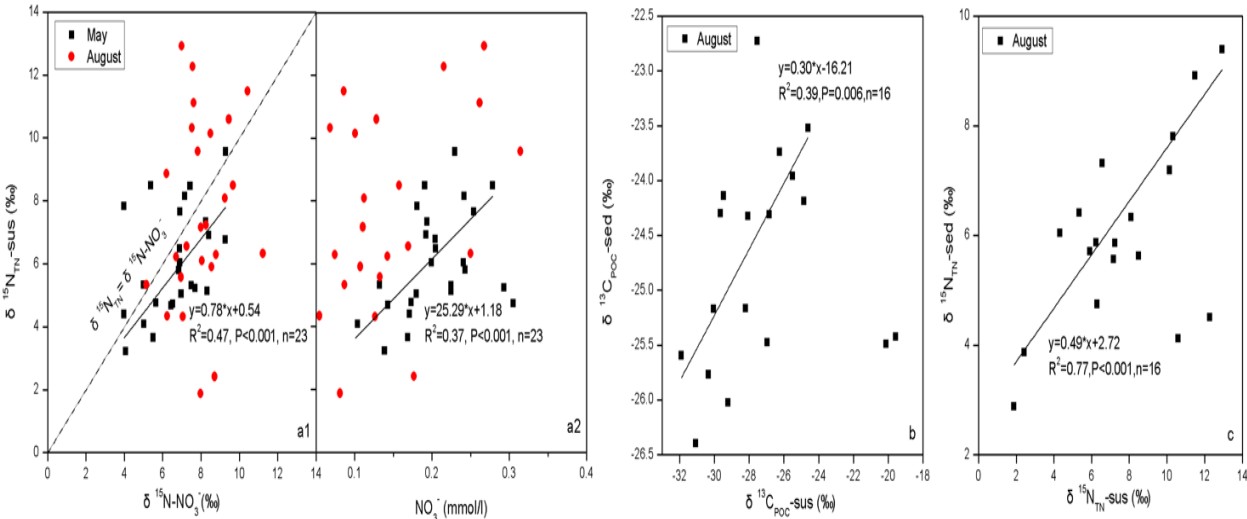

Fig. 7 (a) Correlations of $\delta^{15}N_{TN}$ values in suspended matters and $\delta^{15}N\text{-}NO_3^-$ (a1) as well as $NO_3^-$ concentrations (a2) in corresponding river water in the Wujiang River; (b) Correlations of $\delta^{13}C$ values between suspended matters (sus) and surface sediments (sed); (c) Correlations of $\delta^{15}N$ values between suspended matters and surface sediments.




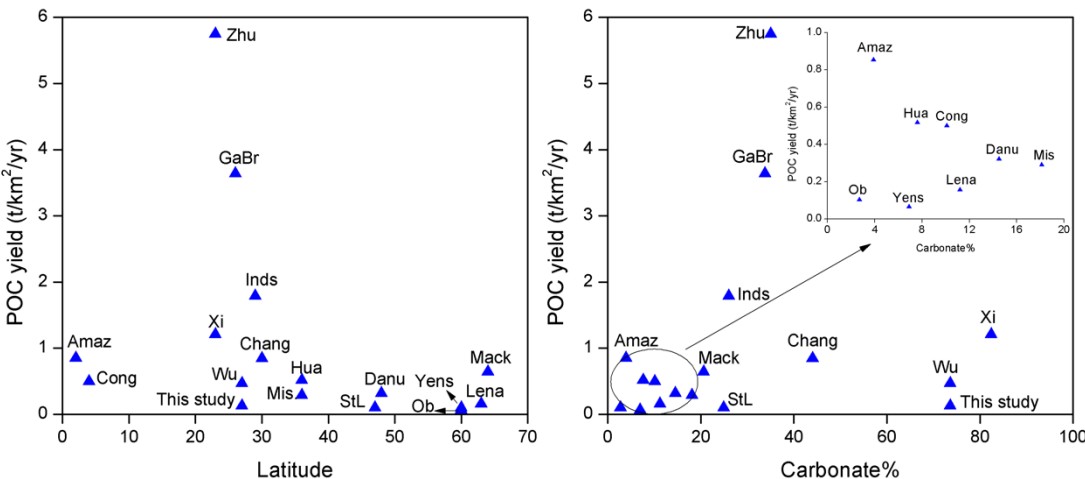

Fig. 8 (a) Relationship between POC yield and latitude; (b) Relationship between POC yield and percentage of carbonate area (carbonate%). Latitude and percentage of carbonate area were taken from Cat et al. (2008) and Amiotte Suchet et al. (2003), respectively.