# Peer review of "Source and flux of POC in a karstic area in the Changjiang River watershed: impacts of reservoirs and extreme drought"

_Biogeosciences, 2015_

## Referee Comment (RC1) · Anonymous Referee #2 · 12 Apr 2016

**Review of the manuscript "Source and flux of POC in a karstic area in the Changjiang River watershed: impacts of reservoirs and extreme drought" by Li et al.**

The paper reports elemental and isotopic data for organic carbon and nitrogen in suspended matter and sediments of the Wujiang River, an important tributary of the Changjiang River that flows into the Three Gorges Reservoir. The Wujiang River it self is affected by a cascade of reservoirs. The objective of the study was to identify the sources of organic matter in the Wujiang River, examine the impact of reservoir and calculate POC fluxes of the Wujiang River. One studied period was high flow (May) and the other in August, during summer drought. Although the results might be relevant to understand the impact of dams on organic matter drained to Three Gorges Reservoir and Changjiang River, it is hard to follow the argumentation and the characterization of organic matter sources at this stage. The paper thus needs drastic re-organization before any re-submission.

(1) One objective of the paper is to study the impact of dams on the organic matter carried or settled in the Wujiang. However, it is not easy to identify how dams affect sampling parameters. Fig. 4 is important but not easy to read. I suggest a diagram comparing the quantitative variations of studied parameters along the river course (as a function of distance) for the two studied periods with the position of dams marked. The points that are considered as directly affected by reservoirs could be clearly identified on Figs. 4, 8, not only in Table 2. It might be more realistic to distinguish points that directly affected by reservoirs and those less affected, rather than "affected" and "unaffected" points. All points are probably more or less affected by the cascade of reservoirs.

(2) The authors used combined DIC $\delta^{13}$C, C/N and $\delta^{15}$N results to identify the source of organic matter.
As shown by the diagram of Fig. 5, there are more than two possible sources. It is thus not clear how the authors made simple quantitative mixing models between phytoplankton and C3 plants, and between C3 and C4 plants on the basis of $\delta^{13}$C alone (results shown on Fig. 6). Most $\delta^{13}$C in Fig. 5 are consistent with a dominant C3 plant source (after given into account the variability of the C3 plant source). The most enriched points most possibly reflect C4 soil plant input and the most depleted one phytoplankton input. It is however not possible to make quantitative estimations (on the basis of $\delta^{13}$C alone) as three possible sources are mixed.
The identification of the phytoplankton end-member in the text is confusing. It is stated that it can be measured on the basis of dissolved DIC $\delta^{13}$C and fractionation factor of -21‰(page 6, lines 10-11). A calculated range (?) of -32,6 to -24,4‰ was given although not DIC $\delta^{13}$C have been given. They could be supplied as supplementary material if available. It is also stated that phytoplankton $\delta^{13}$C is lower then 30‰ (page 6, line 5), then that it has a typical range between -42 and -24‰ (page 6 line 13).
An average $\delta^{13}$C of -13,4‰ is given for C4 plants in the catchment from Tao et al. (2009) (page 6, line 24), but the sigma value (with reference) is not given. The exact values and references (published in English) for the average and sigma values of C3 fresh plant and soil end-members (shown in Fig. 5) were not given. Note that the average $\delta^{13}$C values for C3 plants (ca. -28‰ from Fig. 5) seems a bit more depleted than expected. If

measurements exist for the main C3 plants in the catchment are available, they could be added as supplementary material.

Fig. 5 clearly shows a set of points with high C/N, suggesting an important contribution of fresh terrestrial plant material, essentially from C3 plants. This point is not discussed.

(3) The discussion on sediments $\delta^{13}C$ is not easy to read.

As shown by the authors (Fig 5, 8 and page 7 lines 10-19), the sediments are enriched in $^{13}C$ (relative to suspended sediments). The authors proposed that there is a relative increase in C4 plant debris in the sediment or preferential loss of light isotopes in the sediment (lines 13-4) and then later proposed a preferential biodegradation of the phytoplankton in the water column (lines 16-17). These three possible options are not discussed.

The $\delta^{13}C$ sediment/suspended sediment plot was introduced later (page 8, lines 14-15) and can be useful in that part of the discussion.

(4) It is not clear why the positive relation between POC and TN (total nitrogen) suggested that a fraction of nitrogen is inorganic (page 5 line 24; page 7 line 5-6). One would expect indeed a positive relation between POC and particulate organic nitrogen, with the slope depending on organic C/N ratio. It could also be useful to specify the possible inorganic forms of nitrogen in sediments and suspended matter.

$\delta^{15}N$ is considered as a tracer of POC source throughout the text (see page 5, line 19 among others). It is actually a tracer of nitrogen source and by consequence of organic matter source.

(5) The discussion of $\delta^{15}N$ is confusing (page 8, lines 1-10).

To explain the variation in $\delta^{15}N$ in suspended matter, the authors refer to dissolved nitrate $\delta^{15}N$ (Fig. 8a). These data are however not given in Table 1.

They used these data to assess that high $\delta^{15}N$ of N in suspended sediments indicated manure and domestic sewage (page 8, lines 1-2), but then to confirm nitrogen input from phytoplankton (line 4-10). The importance of sewage organic matter / phytoplankton N derived from sewage- nitrate is not at all discussed.

The authors stated that the enrichment on $^{15}N$ of organic nitrogen in dam-affected reservoir is consistent with increased input of phytoplankton nitrogen. However this enrichment does seem significant (7.99±4,12 / 7.42±2.49).

Furthermore, the good correlation between $^{15}N$ in sediment and suspended matter (Fig. 8c) is not really discussed. Relative high $\delta^{15}N$ values are observed in both the sediment and suspended sediment. This is not in agreement with previous assumptions made by the authors that high $\delta^{15}N$ is essentially tied to the phytoplankton input and that phytoplankton is mainly decomposed in the water column. This might suggest an enriched source of "recalcitrant" N or an incorporation of phytoplankton-N in recalcitrant sediment nitrogen.

(5) Figures (3, 9 and may be 7) and tables (1, 3, 5) might be supplied as supplementary materials. The information from table 2 can be given in the text. It is better to put the measurements for a given site on one given line in Table 1. For Fig. 6, see above point 2.

(6) I suggest a revision of the paper by native English speaker.

*Minor comments*

page 1, line 27 "characterized" instead of charactered

page 1, line 30 "tracer of particulate organic matter " instead of "tracer of POC"

page 2, line 6-5 "Moreira-Turcq" instead of "Moreira", "Aucour" instead of "Ancouret"

page 2, line 8 "flows into" instead of "empties"

page 2, line 29 "whereas" instead ".Whereas"

page 3, lines 12-13 ; It is not clear how freeze-dried filters with suspended sediments could be separated to calculate the mass of suspended matter and sieved to 200 mesh.

page 3, line 20 "where" instead of "Where"

page 3, line 18 The measurements have been made at the Chinese Academy of Forestry. The name, location of laboratory coudl be given as it does not appear in affiliations.

Page 3 line 30 "sedimentation" instead of "precipitation"

Page 4 line 3 "Elemental and isotope composition" instead of "composition of element and isotope"

Page 4 line 8 and throughout the text "cascade of reservoirs" instead of "cascade reservoirs"

---

## Author Comment (AC1) · 13 Apr 2016

Comment 1: One objective of the paper is to study the impact of dams on the organic matter carried or settled in the Wujiang. However, it is not easy to identify how dams affect sampling parameters. Fig. 4 is important but not easy to read. I suggest a diagram comparing the quantitative variations of studied parameters along the river course (as a function of distance) for the two studied periods with the position of dams marked. The points that are considered as directly affected by reservoirs could be clearly identified on Figs. 4, 8, not only in Table 2. It might be more realistic to distinguish points that directly affected by reservoirs and those less affected, rather than "affected" and "unaffected" points. All points are probably more or less affected by the

cascade of reservoirs. Response: Thank you for your valuable comment. According to the suggestion, we tried to make a diagram with quantitative variations of studied parameters along the river course (as a function of distance). However, this diagram was not easy to read because there were many dams and sampling sites. Moreover, some sites were too close to present them clearly. In the Fig. 4, the dams were marked in order to make it easy to read. According to the comment, we have modified the description of sampling sites as "directly affected by reservoirs" and "less affected by reservoirs".

Comment 2: The authors used combined DIC $\delta13C$, C/N and $\delta15N$ results to identify the source of organic matter. âŠă As shown by the diagram of Fig. 5, there are more than two possible sources. It is thus not clear how the authors made simple quantitative mixing models between phytoplankton and C3 plants, and between C3 and C4 plants on the basis of $\delta13C$ alone (results shown on Fig. 6). Most $\delta13C$ in Fig. 5 are consistent with a dominant C3 plant source (after given into account the variability of the C3 plant source). The most enriched points most possibly reflect C4 soil plant input and the most depleted one phytoplankton input. It is however not possible to make quantitative estimations (on the basis of $\delta13C$ alone) as three possible sources are mixed. Response: Thank you for your valuable comment. In the present study, the linear relationship of TN and POC was relatively weak compared with other studies (see details in answer to comment 4). This could limit the usefulness of C/N ratios as a tracer of particulate organic matter source. The $\delta13C$ of POC in the suspend matters in August averaged -27.23±2.93‰ indicating that the terrestrial source was a major source of POC. While the corresponding C/N averaged 8.84±3.73, indicating that phytoplankton was a dominant source of POC. The lack of power to resolve the source of organic matter using C/N ratios was also noticed in other studies (Sarma et al., 2012; Middelburg and Herman 2007). Thus, we use $\delta13C$ to calculate the contribution of different sources of organic matter. Soil organic matter (including litterfall) is eventually a mixture of residues from the overlying vegetation, which is composed of C3 and C4 plants. Thus, the $\delta13C$ values of soil organic matter can be used to reflect the terrestrial sources of POC. In the present study, the contribution of C3 plants and C3 plant-dominated soil together represented the C3 source; C4 plants and C4 plant-dominated soil together represented the C4 source. According to the source criteria developed from the $\delta$13C, we think that the contribution of phytoplankton, C3 and C4 source can be distinguished. Middelburg, J.J., Herman, P.M.J.: Organic matter processing in tidal estuaries, Mar. Chem. 106:127–147, 2007.

âŚą The identification of the phytoplankton end-member in the text is confusing. It is stated that it can be measured on the basis of dissolved DIC $\delta$13C and fractionation factor of -21%(page 6, lines 10-11). A calculated range (?) of -32.6 to -24.4‰ was given although not DIC $\delta$13C have been given. They could be supplied as supplementary material if available. It is also stated that phytoplankton $\delta$13C is lower than -30‰ (page 6, line 5), then that it has a typical range between -42 and -24‰ (page 6 line 13). Response: Thank you for your valuable comment. Measured $\delta$13C-DIC in the Wujiang River ranged from $-11.55$‰ to $-3.41$‰ with an average value of -8.67‰These data is included in another paper, which is under review. Thus, we do not show them in Table 1. Based on the range of $\delta$13C-DIC and fractionation factor of -21%, the estimated $\delta$13C values for phytoplankton ( $-32.6$‰ $\sim$ $-24.4$‰ can be obtained. In order to make it uniform, the typical $\delta$13C range of phytoplankton from Mook and Tan (1991) was corrected as Kendall et al. (2001) and references therein based on the study by Li (2009) on $\delta$13C of phytoplankton (-29.5$\pm$5.5‰ in Maotiao River (a tributary of Wujiang River).

âŚćAn average $\delta$13C of -13.4‰ is given for C4 plants in the catchment from Tao et al. (2009) (page 6, line 24), but the sigma value (with reference) is not given. The exact values and references (published in English) for the average and sigma values of C3 fresh plant and soil end-members (shown in Fig. 5) were not given. Note that the average $\delta$13C values for C3 plants (ca. -28‰ from Fig. 5) seem a bit more depleted than expected. If measurements exist for the main C3 plants in the catchment are available, they could be added as supplementary material. Response: Thank you for

your valuable comment. The δ13C of different endmembers were taken from other studies in the Wujiang River (Tao et al., 2009 and references therein; Li 2009; Wu et al., 2007). Unfortunately, we did not collect the different endmembers of organic matter in the Wujiang River. Thus, the related data were not shown in supplementary material. The δ13C of different endmembers (mean ± standard deviation) were added in Fig. 4 (new edition).

âŠč Fig. 5 clearly shows a set of points with high C/N, suggesting an important contribution of fresh terrestrial plant material, essentially from C3 plants. This point is not discussed. Response: Thank you for your valuable comment. As shown by the contribution of different organic matter, POC in the Wujiang River was mainly derived from the terrestrial source. Given the limitation of C/N in the studied basin (see details in answer to comment 4), it was difficult to distinguish the contribution of C3 plants from the C3 plant-dominated soil. In the present study, these two sources represented C3 source.

Comment 3: The discussion on sediments δ13C is not easy to read. As shown by the authors (Fig 5, 8 and page 7 lines 10-19), the sediments are enriched in 13C (relative to suspended sediments). The authors proposed that there is a relative increase in C4 plant debris in the sediment or preferential loss of light isotopes in the sediment (lines 13-4) and then later proposed a preferential biodegradation of the phytoplankton in the water column (lines 16-17). These three possible options are not discussed. The δ13C sediment/suspended sediment plot was introduced later (page 8, lines 14-15) and can be useful in that part of the discussion. Response: Thank you for your valuable comment. As mentioned in the comment, the enriched δ13C in the sediments might be attributed to three causes. Given that POC and TN contents were higher in most sediment samples than suspended sediments, we think that the biodegradation of the phytoplankton was not significant. Thus, the higher δ13C in the sediments was mainly due to the contribution of refractory allochthonous organic matter (i.e. C4 plants). The related discussion has been added in the corresponding section.

Comment 4: It is not clear why the positive relation between POC and TN (total nitrogen) suggested that a fraction of nitrogen is inorganic (page 5 line 24; page 7 line 5-6). One would expect indeed a positive relation between POC and particulate organic nitrogen, with the slope depending on organic C/N ratio. It could also be useful to specify the possible inorganic forms of nitrogen in sediments and suspended matter. $\delta$15N is considered as a tracer of POC source throughout the text (see page 5, line 19 among others). It is actually a tracer of nitrogen source and by consequence of organic matter source. Response: Thank you for your constructive comment. Ratios of C/N have been used to distinguish sources of organic carbon in marine and coastal environments based on the assumption that all of the sedimentary TN exclusively reflects N bound to organic matter (Meyers, 1997). As mentioned in the comment, the slope of linear relationship between TN and POC content depend on organic C/N ratio and the intercept value could reflect the inorganic nitrogen. In the present study, the linear relationship of TN and POC was relatively weak (May: TN=0.07*POC+0.09, R2=0.54, P<0.001; August: TN=0.04*POC+0.23, R2=0.39, P<0.001) compared with other studies (R2=0.71 in Hu et al., 2006; R2=0.9 in Guerra et al., 2013). Thus we think that the inorganic nitrogen in the present study was relatively high in comparison with the above studies. The related discussion has been added in this part. The related reference: Meyers, P.A., 1997. Organic geochemical proxies of paleoceanographic, pleolimnologic, and paleoclimatic processes. Organic Geochemistry 27, 213-250

Comment 5: The discussion of $\delta$15N is confusing (page 8, lines 1-10). âŠă To explain the variation in $\delta$15N in suspended matter, the authors refer to dissolved nitrate $\delta$15N (Fig. 8a). These data are however not given in Table 1. They used these data to assess that high $\delta$15N of N in suspended sediments indicated manure and domestic sewage (page 8, lines 1-2), but then to confirm nitrogen input from phytoplankton (line 4-10). The importance of sewage organic matter / phytoplankton N derived from sewage-nitrate is not at all discussed. Response: Thank you for your valuable comment. The dual isotopes of dissolved nitrate are included in another paper which is under review. Thus, we do not show them in Table 1. The discussion about anthropogenic source

has been rewritten.

âŚą...... the good correlation between 15N in sediment and suspended matter (Fig. 8c) is not really discussed. Relative high $\delta$15N values are observed in both the sediment and suspended sediment. This is not in agreement with previous assumptions made by the authors that high $\delta$15N is essentially tied to the phytoplankton input and that phytoplankton is mainly decomposed in the water column. This might suggest an enriched source of "recalcitrant" N or an incorporation of phytoplankton-N in recalcitrant sediment nitrogen. Response: Thank you for your constructive comment. The discussion about the correlation of sediment and suspended matter has been rewritten in the corresponding section.

Comment 6: Figures (3, 9 and may be 7) and tables (1, 3, 5) might be supplied as supplementary materials. The information from table 2 can be given in the text. It is better to put the measurements for a given site on one given line in Table 1. For Fig. 6, see above point 2. Response: Thank you for your valuable advice. As suggested by the reviewer, Figure 3, table (1, 3, 5) were put in the supplementary materials. Considering that the Figure 7 and Figure 9 are meaningful for comparison with the world rivers, we put them in the paper. In order to make it easy to understand the comparison of parameters between dam-affected sites and less dam-affected sites, the information in Table 2 was shown as a table.

Comment 7: I suggest a revision of the paper by native English speaker. page 1, line 27 "characterized" instead of "charactered" ...... Page 4 line 8 and throughout the text "cascade of reservoirs" instead of "cascade reservoirs" Response: Thank you for careful work. We have accepted the suggestion and made corresponding corrections according to the comment.

———————————————

**Response to Comments from Referee #2**

**Comment 1**: One objective of the paper is to study the impact of dams on the organic matter carried or settled in the Wujiang. However, it is not easy to identify how dams affect sampling parameters. Fig. 4 is important but not easy to read. I suggest a diagram comparing the quantitative variations of studied parameters along the river course (as a function of distance) for the two studied periods with the position of dams marked. The points that are considered as directly affected by reservoirs could be clearly identified on Figs. 4, 8, not only in Table 2. It might be more realistic to distinguish points that directly affected by reservoirs and those less affected, rather than "affected" and "unaffected" points. All points are probably more or less affected by the cascade of reservoirs.

**Response:** Thank you for your valuable comment. According to the suggestion, we tried to make a diagram with quantitative variations of studied parameters along the river course (as a function of distance). However, this diagram was not easy to read because there were many dams and sampling sites. Moreover, some sites were too close to present them clearly. In the Fig. 4, the dams were marked in order to make it easy to read. According to the comment, we have modified the description of sampling sites as "directly affected by reservoirs" and "less affected by reservoirs".

**Comment 2**: The authors used combined DIC $\delta^{13}C$, C/N and $\delta^{15}N$ results to identify the source of organic matter.

①As shown by the diagram of Fig. 5, there are more than two possible sources. It is thus not clear how the authors made simple quantitative mixing models between phytoplankton and C3 plants, and between C3 and C4 plants on the basis of $\delta^{13}C$ alone (results shown on Fig. 6). Most $\delta^{13}C$ in Fig. 5 are consistent with a dominant C3 plant source (after given into account the variability of the C3 plant source). The most enriched points most possibly reflect C4 soil plant input and the most depleted one phytoplankton input. It is however not possible to make quantitative estimations (on the basis of $\delta^{13}C$ alone) as three possible sources are mixed.

**Response:** Thank you for your valuable comment. In the present study, the linear relationship of

**Fig. 1.**

---

## Referee Comment (RC2) · Anonymous Referee #3 · 18 Apr 2016

Comments on "Source and flux of POC in a karstic area in the Changjiang River watershed: impacts of reservoirs and extreme drought" authored by Ji et al.

The authors measured δ13CPOC, δ15NTN and C/N ratios in both suspended and surface
Sediments along the Wujiang River and attempted to identify source and flux of POC in the    Wujiang River and addressed the impacts of reservoir on POC flux into the Changjiang (or finally into the East China Sea). I think the authors had lots of data sets in two different seasons, but the whole paper presentation is not very good. I am confused about the title , introduction and interpretation about the content.    Overall, I think the paper need a major revision before it can be considered to be published. My major comments are as follows:

1. The title is not suitable because the data set of POC flux and POC sources in the Changjiang River is only from upper branch. The authors mentioned POC fluxes in different rivers in the introduction, but it did not touch real POC flux in the Changjiang River mouth or the East China Sea. The title should be modified.

2. As addressed above, the introduction described the importance of riverine POC flux to different marginal seas and the main objective of the manuscript seems to emphasize the effect of Three Gorge Dam on POC flux to the East China Sea. I suggest that the authors should review possible difference of POC flux in the Changjiang before and after the construction of Three Gorge Dam.    For example, the authors keeping saying POC flux to marginal seas are quite important, but they said that….. "Wujiang River is still scarce after the Three Gorges Dam began impounding sediment in 2004. Based on analyses of δ13CPOC, δ15NTN and C/N ratios in the suspended and surface sediments, this study identified source and flux of POC in the Wujiang River and examined the impacts of reservoir and climate." I did not see the description above associated with whole Changjiang watershed because the Wujiang River is only a part of Changjiang branches. Plus, they attempted to study the impacts of reservoir (Three Gorge Dam?) and climate based on two season data sets. I think the little data can not wholly support their perspective. Instead, the author should point out what POC flux in the Changjiang River before the construction of Three Gorge Dam are in the introduction ? In the next step, they want to examine the impacts of trapped POC in Three Gorge Dam affecting the output of Changjiang River. Anyway, the introduction and abstract need to be re-worked. A useful reference should be helpful for the authors. Hung et al. (2003). Fluxes of particulate organic carbon in the East China Sea in summer. BG, 10, 6469-6484.

3.  Source of organic carbon in suspended particles and sediments are roughly separated to two main sources which may not be right. I can see authors discussed the percentage of each compound (C3 and phytoplankton) in equations 1~3 in the text, but they also explain possible sources such as C4 and C4-soil and include these compounds into equations. It is quite inconsistent for the data interpretation. I suggest the authors need do it based on other sources.

Other comments

Results.

3.1 line 25-26, it has been described in the method, delete it.

3.2 Line 21-23 content should show in the method section

Line 27, how significant? Showing p and n

Discussion

4.1 ….line 23 suggested the dominant terrestrial contribution to SPM in May and increased phytoplankton input in August. As discussed, all samples were collected in the fresh water suspended particles or sediments, it is absolutely from terrestrial source.

4.2 line 27-29 showed a relatively significant positive correlation, which suggested that a fraction of TN was inorganic nitrogen in the SPM. Why? Thus, the phytoplankton inputs might be overestimated based on C/N ratios. How can you explain this? Is it related to Redfield ratio?

P6, line9-20, are C3 and phytoplankton POC only two sources? How about other sources? Do authors have other C sources like C4 etc. ? If other C sources exist, the equations 2 and need to solved? There is a useful reference (Hung et al., ECSS, 84, 566-572) which reported that POC/Chl-a ratio in summer ranged from 50 to 70, if the authors have Chlorophyll-a data. They can estimate POC source from phytoplankton based on suspended POC data.

Line 25-34 why the phytoplankton in affected and the unaffected areas has large difference? They are both affected by fresh water largely. Is it due to residence time or other carbon sources?

P7, …Compared with SPM, the elevated C/N ratios of surface sediments indicated more land-derived fraction contribution to the surface sediments. What other sources contributed to POC in sediments ? Line 10-13, If C4 is partially associated with POC, then the end member mixing model should be modified.

4.3 & 4.4 Flux of POC in Wujiang River, as mentioned early, the amount of POC flux is totally into Three Gorge Dam? It is quite simple to estimate POC and PIC fluxes.

The important thing should be focused on how much POC are trapped in the TGD and affect the POC export flux to the East China Sea. I think this portion should need deep discussion. For example, the author should compare the POC flux at the upper and lower watershed of TGD before and after construction of TGD. Plus, the authors keep saying possible impacts of the TGD, …the variations of suspended sediment load could reflect the POC flux variations under the condition of dam and extreme drought….” What my understanding is that the authors should provide POC flux in the lower watershed of TGD rather than upper watershed because these upper POC finally will empty TGD, right?

Figure 2, the authors should provide water discharge data in the lower wathershed of Changjiang such as Datong station and compare what is the difference of water discharge and POC flux between flood and drought seasons. If the authors have those data sets, the manuscript will provide evidence if TGD has significant impact or not. Figure 5, there are five carbon components in the figure showing different contributions of carbon sources to suspended and/or sediments. However, the authors only used two end-member to calculate possible contributions of phytoplankton and C3-plant. Why?

---

## Author Comment (AC2) · 5 May 2016

General overview: The authors measured $\delta13CPOC$, $\delta15NTN$ and C/N ratios in both suspended and surface sediments along the Wujiang River and attempted to identify source and flux of POC in the Wujiang River and addressed the impacts of reservoir on POC flux into the Changjiang (or finally into the East China Sea). I think the authors had lots of data sets in two different seasons, but the whole paper presentation is not very good. I am confused about the title, introduction and interpretation about the content. Overall, I think the paper need a major revision before it can be considered to be published.

Response: Thank you very much for reviewing the manuscript and for the valuable

comments. We have revised the manuscript based on the comments and suggestions.

My major comments are as follows: Comment 1. The title is not suitable because the data set of POC flux and POC sources in the Changjiang River is only from upper branch. The authors mentioned POC fluxes in different rivers in the introduction, but it did not touch real POC flux in the Changjiang River mouth or the East China Sea. The title should be modified.

Response: Thank you for the comment. The Wujiang River is the largest tributary of the upper Changjiang River in its south bank. Although there are many tributaries for Changjiang River, Wujiang River is a typical karst watershed. Thus, we think that the title can represent the major objectives of our study.

Comment 2. As addressed above, the introduction described the importance of riverine POC flux to different marginal seas and the main objective of the manuscript seems to emphasize the effect of Three Gorge Dam on POC flux to the East China Sea. I suggest that the authors should review possible difference of POC flux in the Changjiang before and after the construction of Three Gorge Dam. For example, the authors keeping saying POC flux to marginal seas are quite important, but they said that. . ... "Wujiang River is still scarce after the Three Gorges Dam began impounding sediment in 2004. Based on analyses of $\delta13CPOC$, $\delta15NTN$ and C/N ratios in the suspended and surface sediments, this study identified source and flux of POC in the Wujiang River and examined the impacts of reservoir and climate." I did not see the description above associated with whole Changjiang watershed because the Wujiang River is only a part of Changjiang branches. Plus, they attempted to study the impacts of reservoir (Three Gorge Dam?) and climate based on two season data sets. I think the little data can not wholly support their perspective. Instead, the author should point out what POC flux in the Changjiang River before the construction of Three Gorge Dam are in the introduction? In the next step, they want to examine the impacts of trapped POC in Three Gorge Dam affecting the output of Changjiang River. Anyway, the introduction and abstract need to be re-worked. A useful reference should be helpful for the authors. Hung et al. (2003). Fluxes of particulate organic carbon in the East China Sea in summer. BG, 10, 6469-6484.

Response: Thank you for the comment. As suggested in the comment, variations of POC flux in the Changjiang River before and after the construction of Three Gorges Dam are important to identify the influence of damming on the local carbon cycle and even on the global carbon cycle to some extent. The related review has been added in the introduction. Eleven artificial dams have been constructed along the mainstream of Wujiang River (Fig. 1). However, related study on the impacts of these cascades of dams is limited. Thus, one of the objectives of our study is to estimate the impacts of the above eleven cascades of dams on the POC source and flux in a karstic watershed. The objectives of this study in the introduction have been made clearer. Two season samples were collected in the present study. These data may lead to a high level of error when estimating the impacts of reservoirs and drought of 2013. However, we think that it is helpful for understanding the variations of POC source and flux in the Wujiang River in the special drought year of 2013.

Comment 3. Source of organic carbon in suspended particles and sediments are roughly separated to two main sources which may not be right. I can see authors discussed the percentage of each compound (C3 and phytoplankton) in equations 1~3 in the text, but they also explain possible sources such as C4 and C4-soil and include these compounds into equations. It is quite inconsistent for the data interpretation. I suggest the authors need do it based on other sources.

Response: Thank you for the comment. According to the comment, we have carefully modified the mixing model using indicators of $\delta$13C values and C/N ratios. The combination of $\delta$13C values and C/N ratios is also used in other studies (Jiang and Ji, 2013; Lu et al., 2013). Lu, F. Y., Liu, Z. Q., Ji, H. B.: Carbon and nitrogen isotopes analysis and sources of organic matter in the upper reaches of the Chaobai River near Beijing, China.ÂăScience China Earth Science,Âă56(2), 217-227, 2013. Jiang, Y. and Ji, H.: Isotopic indicators of source and fate of particulate organic carbon in a karstic

watershed on the Yunnan-Guizhou Plateau. Appl. Geochem., 36, 153−167, 2013.

Results. Comment 3.1 line 25-26, it has been descripted in the method, delete it. Response: Thank you for the comment. The mentioned description has been deleted. Comment 3.2 Line 21-23 content should show in the method section Line 27, how significant? Showing p and n Response: Thank you for the comment. The mentioned content (Line 21-23) has been moved in the method section. The values of relation coefficient and p have been added in the Table S2.

Discussion Comment 4.1 . . ..line 23 suggested the dominant terrestrial contribution to SPM in May and increased phytoplankton input in August. As discussed, all samples were collected in the fresh water suspended particles or sediments, it is absolutely from terrestrial source.

Response: Thank you for the comment. As mentioned in the comment, POC is generally derived from terrestrial source in the fresh water. However, aquatic source increases as more and more artificial dams are constructed. A similar study can be seen from one tributary of Wujiang River (Jiang and Ji, 2013), in which POC of SPM was mainly derived from phytoplankton.

Comment 4.2 line 27-29 showed a relatively significant positive correlation, which suggested that a fraction of TN was inorganic nitrogen in the SPM. Why? Thus, the phytoplankton inputs might be overestimated based on C/N ratios. How can you explain this? Is it related to Redfield ratio?

Response: Thank you for the valuable comment. According to Meyers (1997), ratios of C/N are used to distinguish sources of organic carbon in marine and coastal environments based on the assumption that all of the sedimentary TN exclusively reflects N bound to organic matter. As discussed in the manuscript, the slope of linear relationship between TN and POC content depend on organic C/N ratio and the intercept value could reflect the inorganic nitrogen. In the present study, the linear relationship of TN and POC was relatively weak (May: TN=0.07*POC+0.09, $R^2$=0.54, P<0.001; August:

TN=0.04*POC+0.23, R2=0.39, P<0.001) compared with other studies (R2=0.71 in Hu et al., 2006; R2=0.9 in Guerra et al., 2013). The intercept of the above regressions was more than zero, which suggested that a fraction of TN was inorganic nitrogen in the SPM (Guerra et al., 2013; Hu et al., 2006). Because contents of total nitrogen included some inorganic nitrogen in the study area, measured C/N ratios were underestimated, which led to phytoplankton inputs overestimated based on measured C/N ratios. Guerra, R., Pistocchi, R. and Vanucci, S.: Dynamics and sources of organic carbon in suspended particulate matter and sediments in Pialassa Baiona lagoon (NW Adriatic Sea, Italy), Estuar. Coast. Shelf S., 135, 24-32, 2013. Hu, J., Peng, P. A., Jia, G., Mai, B. and Zhang, G.: Distribution and sources of organic carbon, nitrogen and their isotopes in sediments of the subtropical Pearl River estuary and adjacent shelf, Southern China, Mar. Chem., 98(2), 274−285, 2006. Meyers, P.A.: Organic geochemical proxies of paleoceanographic, pleolimnologic, and paleoclimatic processes. Organic Geochemistry 27, 213-250, 1997.

Comment P6, line 9-20, are C3 and phytoplankton POC only two sources? How about other sources? Do authors have other C sources like C4 etc.? If other C sources exist, the equations 2 and need to solved? There is a useful reference (Hung et al., ECSS, 84, 566-572) which reported that POC/Chl-a ratio in summer ranged from 50 to 70, if the authors have Chlorophyll-a data. They can estimate POC source from phytoplankton based on suspended POC data.

Response: Thank you for constructive comment. The method in the study of Huang et al. (2003) is useful to estimate POC source from phytoplankton. Unfortunately, we did not measure the values of Chlorophyll-a in our study. As mentioned in the comment, there exist C4 source in addition to C3 and phytoplankton. The mixing model of endmembers was modified.

Comment Line 25-34 why the phytoplankton in affected and the unaffected areas has large difference? They are both affected by fresh water largely. Is it due to residence time or other carbon sources?

Response: Thank you for the comment. Two mechanisms could explain the elevated phytoplankton contribution in sites affected by reservoirs: (1) extended water retention time in reservoirs with low flow; (2) increasing light availability due to the low TSS concentrations in reservoirs. This related discussion was included in the section 4.4 (Impacts of reservoir and climate on riverine POC).

Comment P7, …Compared with SPM, the elevated C/N ratios of surface sediments indicated more land-derived fraction contribution to the surface sediments. What other sources contributed to POC in sediments? Line 10-13, If C4 is partially associated with POC, then the end member mixing model should be modified.

Response: Thank you for the valuable comment. According to the comment, we have carefully modified the mixing model using indicators of $\delta$13C values and C/N ratios.

Comment 4.3 & 4.4 Flux of POC in Wujiang River, as mentioned early, the amount of POC flux is totally into Three Gorge Dam? It is quite simple to estimate POC and PIC fluxes. The important thing should be focused on how much POC are trapped in the TGD and affect the POC export flux to the East China Sea. I think this portion should need deep discussion. For example, the author should compare the POC flux at the upper and lower watershed of TGD before and after construction of TGD. Plus, the authors keep saying possible impacts of the TGD, …the variations of suspended sediment load could reflect the POC flux variations under the condition of dam and extreme drought….." What my understanding is that the authors should provide POC flux in the lower watershed of TGD rather than upper watershed because these upper POC finally will empty TGD, right?

Response: Thank you for the comment. The Wujiang River flows into the Three Gorges Reservoir in Chongqing Municipality. It is better to estimate the POC flux using a depth-integrated concentration (Coynel et al., 2005). However, due to the large elevation gradients with about 1500 m in its upper reach and 500 m in its lower reach, Wujiang River has high flow rates. This makes it difficult to collect samples in different water

depths. The POC concentration of river mouth is used to calculate the POC flux, which is frequently used in other studies (Aucour et al., 2006; Tao et al., 2009). As mentioned in the comment, it is very important to study the influence of TGD on POC export flux to the East China Sea. For our study area, there are eleven cascades of reservoirs along the mainstream of Wujiang River. The objective of our study is to examine the impacts of these cascades of reservoirs. The impact of TGD on the POC export to East China Sea is not our goal. Because suspended sediment at the mouth of Wujiang River directly flowed into Three Gorges Reservoir (TGR), the impact of climate on TGR sediment revealed the similar impacts on the mouth of Wujiang River. Thus, in order to estimate the impacts of climate on Wujiang River, we compare sediments inputs in the upper watershed of TGR between normal and drought year. Coynel, A., Seyler, P., Etcheber, H., Meybeck, M. and Orange, D.: Spatial and seasonal dynamics of total suspended sediment and organic carbon species in the Congo River, Global Biogeochem. Cy., 19, doi:10.1029/2004GB002335, 2005. Aucour, A.M., France-Lanord, C., Pedoja, K., Pierson-Wickmann, A.C., and Sheppard, S.M.F.: Fluxes and sources of particulate organic carbon in the Ganga-Brahmaputra river system, Global Biogeochem. Cy., 20, doi:10.1029/2004GB002324, 2006. Tao, F.X., Liu, C.Q. and Li, S.L.: Source and flux of POC in two subtropical karstic tributaries with contrasting land use practice in the Yangtze River Basin, Appl. Geochem., 24 (11), 2102−2112, 2009.

Comment Figure 2, the authors should provide water discharge data in the lower watershed of Changjiang such as Datong station and compare what is the difference of water discharge and POC flux between flood and drought seasons. If the authors have those data sets, the manuscript will provide evidence if TGD has significant impact or not.

Response: Thank you for the comment. We agree with the comment that data in the Datong station is important to estimate the impact of TGD. Unfortunately, we fail to provide POC flux in Datong station. We think that this would not influence our conclusions since our study is to examine the impact of eleven cascades of reservoirs along the

mainstream of Wujiang River on the POC export flux.

Comment Figure 5, there are five carbon components in the figure showing different contributions of carbon sources to suspended and/or sediments. However, the authors only used two end-member to calculate possible contributions of phytoplankton and C3-plant. Why?

Response: Thank you for the comment. According to the comment, the mixing model has been modified by combined use of indicators of $\delta$13C values and C/N ratios.

Please also note the supplement to this comment:
http://www.biogeosciences-discuss.net/bg-2015-655/bg-2015-655-AC2-supplement.pdf

––––––––––––––––––––––––

---

## Author Response (AR2)

Respected Prof. Natascha Töpfer:

On behalf of my co-authors, we thank you very much for giving us an opportunity to revise our manuscript. According to the comments from Prof. Manmohan Sarin, we have made modifications in the following text. We hope that the following amendments or interpretations can obtain a more accurate manuscript.

We hope that the revised manuscript could satisfy the requirements for publication in the esteemed journal. Thank you and the reviewers again.

We are looking forward to the final good news from your esteemed journal.

Sincerely yours,

Ji Hongbing

**Reply to Prof. Manmohan Sarin**
**Hongbing Ji, Cai Li, Huaijian Ding, Yang Gao**
**"Source and flux of POC in a karstic area in the Changjiang River watershed: impacts of reservoirs and extreme drought"**

**General overview:**
Overall, Authors have responded reasonably well to all the comments made by the two Referees. However, "Abstract" is still somewhat confusing and lacks clarity. Authors are suggested to make following changes in the abstract, or as they feel appropriate:
**Response:** Thank you very much for reviewing the manuscript and for the valuable suggestions. We have revised the abstract based on the suggestions.

**Comment 1.** Line 10 should read as: "Isotope data for suspended sediments indicate that POC was mainly derived from .......
**Response:** Thank you for the valuable suggestion. The corresponding changes in the abstract have been made.

**Comment 2.** Line 14, change to: ....... carbon pools are tightly coupled.
**Response:** Thank you for the valuable suggestion. The related words have been revised.

**Comment 3.** Line 14 should read as: Our conservative estimate suggests that $1.17 \times 10^{10}$ g of POC is transported to the Three Gorges Reservoir during the study period in 2013.
**Response:** Thank you for the valuable suggestion. According to the suggestion, we have revised the related sentence.

**Comment 4.** Line 15 change to: …….was much lower than large rivers with high abundance of carbonate minerals.

**Response:** Thank you for the valuable suggestion. The corresponding words have been revised.

**Comment 5.** Line 16 should read as: Based on the distribution pattern of POC yield, it is inferred that carbonate minerals (lithology) do not contribute significantly to the riverine POC.

**Response:** Thank you for the valuable suggestion. According to the suggestion, we have revised the corresponding sentence.

Other change:

Line 20 Page 6 and Line 7 Page 8: "(unpublished data)" has been changed to "(Li and Ji, 2016)".

The values of $\delta^{13}$C-DIC, $\delta^{15}$N-NO$_3^-$ and $\delta^{18}$O-NO$_3^-$ in the Wujiang River are recently published in the study of Li and Ji (2016). Therefore, the related reference is added.